# Learning improved dynamics model in reinforcement learning by incorporating the long term future

**Nan Rosemary Ke**[*][♦], **Amanpreet Singh**[*][♦][‡], **Ahmed Touati**[♥][♦], **Anirudh Goyal**[♥]

**Yoshua Bengio**[♥], **Devi Parikh**[♣][♦] **& Dhruv Batra**[¶][♦]

## Abstract

In model-based reinforcement learning, the agent interleaves between model learning and planning. These two components are inextricably intertwined. If the model is not able to provide sensible long-term prediction, the executed planner would exploit model flaws, which can yield catastrophic failures. This paper focuses on building a model that reasons about the long-term future and demonstrates how to use this for efficient planning and exploration. To this end, we build a latent-variable autoregressive model by leveraging recent ideas in variational inference. We argue that forcing latent variables to carry future information through an auxiliary task substantially improves long-term predictions. Moreover, by planning in the latent space, the planner's solution is ensured to be within regions where the model is valid. An exploration strategy can be devised by searching for unlikely trajectories under the model. Our method achieves higher reward faster compared to baselines on a variety of tasks and environments in both the imitation learning and model-based reinforcement learning settings.

## 1 Introduction

Reinforcement Learning (RL) is an agent-oriented learning paradigm concerned with learning by interacting with an uncertain environment. Combined with deep neural networks as function approximators, deep reinforcement learning (deep RL) algorithms recently allowed us to tackle highly complex tasks. Despite recent success in a variety of challenging environment such as Atari games (Bellemare et al., 2013) and the game of Go (Silver et al., 2016), it is still difficult to apply RL approaches in domains with high dimensional observation-action space and complex dynamics.

Furthermore, most popular RL algorithms are model-free as they directly learn a value function (Mnih et al., 2015) or policy (Schulman et al., 2015; 2017) without trying to model or predict the environment's dynamics. Model-free RL techniques often require large amounts of training data and can be expensive, dangerous or impossibly slow, especially for agents and robots acting in the real world. On the other hand, model-based RL (Sutton, 1991; Deisenroth & Rasmussen, 2011; Chiappa et al., 2017) provides an alternative approach by learning an explicit representation of the underlying environment dynamics. The principal component of model-based methods is to use an estimated model as an internal simulator for planning, hence limiting the need for interaction with the environment. Unfortunately, when the dynamics are complex, it is not trivial to learn models that are accurate enough to later ensure stable and fast learning of a good policy.

The most widely used techniques for model learning are based on one-step prediction. Specifically, given an observation $o_t$ and an action $a_t$ at time $t$, a model is trained to predict the conditional distribution over the immediate next observation $o_{t+1}$, i.e $p(o_{t+1} \mid o_t, a_t)$. Although computationally easy, the one-step prediction error is an inadequate proxy for the downstream performance of model-based methods as it does not account for how the model behaves when com-

♦ Mila, Université de Montréal, ♥ Facebook AI Research, ♣ Polytechnique Montréal, ¶ CIFAR Senior Fellow, ‡ Work done at Facebook AI Research, * Georgia Institute of Technology
Corresponding authors: `rosemary.nan.ke@gmail.com`

posed with itself. In fact, one-step modelling errors can compound after multiple steps and can degrade the policy learning. This is referred to as the compounding error phenomenon (Talvitie, 2014; Asadi et al., 2018; Weber et al., 2017). Other examples of models are autoregressive models such as recurrent neural networks (Mikolov et al., 2010) that factorize naturally as $\log p_\theta(o_{t+1}, a_{t+1}, o_{t+2}, a_{t+2}, \dots \mid o_t, a_t) = \sum_t \log p_\theta(o_{t+1}, a_{t+1} \mid o_1, a_1, \dots o_t, a_t)$. Training autoregressive models using maximum likelihood results in 'teacher-forcing' that breaks the training over one-step decisions. Such sequential models are known to suffer from accumulating errors as observed in (Lamb et al., 2016; Bengio et al., 2015).

Our key motivation is the following – a model of the environment should reason about (*i.e.* be trained to predict) *long-term transition dynamics* $p_\theta(o_{t+1}, a_{t+1}, o_{t+2}, a_{t+2}, \dots \mid o_t, a_t)$ and not just single step transitions $p_\theta(o_{t+1} \mid o_t, a_t)$. That is, the model should predict what will happen in the long-term future, and not just the immediate future. We hypothesize (and test) that such a model would exhibit less cascading of errors and would learn better feature embeddings for improved performance.

One way to capture long-term transition dynamics is to use latent variables recurrent networks. Ideally, latent variables could capture higher level structures in the data and help to reason about long-term transition dynamics. However, in practice it is difficult for latent variables to capture higher level representation in the presence of a strong autoregressive model as shown in Gulrajani et al. (2016); Goyal et al. (2017); Guu et al. (2018). To overcome this difficulty, we leverage recent advances in variational inference. In particular, we make use of the recently proposed Z-forcing idea (Goyal et al., 2017), which uses an auxiliary cost on the latent variable to predict the long-term future. Keeping in mind that more accurate long-term prediction is better for planning, we use two ways to inject future information into latent variables. Firstly, we augment the dynamics model with a backward recurrent network (RNN) such that the approximate posterior of latent variables depends on the summary of future information. Secondly, we force latent variables to predict a summary of the future using an auxiliary cost that acts as a regularizer. Unlike one-step prediction, our approach encourages the predicted future observations to remain grounded in the real observations.

Injection of information about the future can also help in planning as it can be seen as injecting a plan for the future. In stochastic environment dynamics, unfolding the dynamics model may lead to unlikely trajectories due to errors compounding at each step during rollouts.

In this work, we make the following key contributions:

1. We demonstrate that having an auxiliary loss to predict the longer-term future helps in faster imitation learning.
2. We demonstrate that incorporating the latent plan into dynamics model can be used for planning (for example Model Predictive Control) efficiently. We show the performance of the proposed method as compared to existing state of the art RL methods.
3. We empirically observe that using the proposed auxiliary loss could help in finding sub-goals in the partially observable 2D environment.

## 2 PROPOSED MODEL

We consider an agent in the environment that observes at each time step $t$ an observation $o_t$. The execution of a given action $a_t$ causes the environment to transition to a new unobserved state, return a reward and emit an observation at the next time step sampled from $p^\star(o_{t+1}|o_{1:t}, a_{1:t})$ where $o_{1:t}$ and $a_{1:t}$ are the observation and action sequences up to time step $t$. In many domains of interest, the underlying transition dynamics $p^\star$ are not known and the observations are very high-dimensional raw pixel observations. In the following, we will explain our novel proposed approach to learn an accurate environment model that could be used as an internal simulator for planning.

We focus on the task of predicting a future observation-action sequence $(o_{1:T}, a_{1:T})$ given an initial observation $o_0$. We frame this problem as estimating the conditional probability distribution $p(o_{1:T}, a_{1:T}|o_0)$. The latter distribution is modeled by a recurrent neural network with stochastic latent variables $z_{1:T}$. We train the model using variational inference. We introduce an approximate posterior over latent variables. We maximize a regularized form of the Evidence Lower Bound (ELBO). The regularization comes from an auxiliary task we assign to the latent variables.

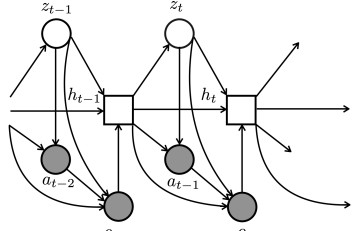 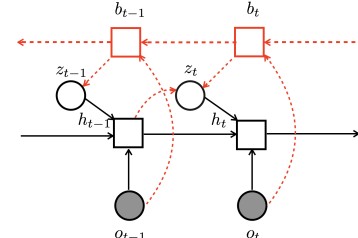

Figure 1: Left: the graphical model representing the generative model $p_\theta$. Right: the architecture of the inference model. The inference network $q_\phi$ uses a backward recurrent state $b_t$ (in red) to approximate the dependence of $z_t$ on future observations. it shares the forward recurrent state $h_{t-1}$ with the generative model to approximate the dependence of $z_t$ on past observations and latent variables. Boxes are deterministic hidden states. circles are random variables and filled circles represent variables observed during training.

## 2.1 GENERATIVE PROCESS

The graphical model in Fig. 1 illustrates the dependencies in our generative model. Observations and latent variables are coupled by using an autoregressive model, the Long Short Term Memory (LSTM) architecture (Hochreiter & Schmidhuber, 1997), which runs through the sequence:

$$h_t = f(o_t, h_{t-1}, z_t) \tag{1}$$

where $f$ is a deterministic non-linear transition function and $h_t$ is the LSTM hidden state at time $t$. According the graphical model in Fig. 1, the predictive distribution factorizes as follows:

$$p_\theta(o_{1:T}, a_{1:T} \mid o_0, h_0) = \int \prod_{t=1}^T \underbrace{p_\theta(o_t \mid a_{t-1}, h_{t-1}, z_t)}_{\text{observation decoder}} \underbrace{p_\theta(a_{t-1} \mid h_{t-1}, z_t)}_{\text{action decoder}} \underbrace{p_\theta(z_t \mid h_{t-1})}_{\text{latent prior}} dz \tag{2}$$

where

1. $p_\theta(o_t \mid a_{t-1}, h_{t-1}, z_t)$ is the observation decoder distribution conditioned on the last action $a_{t-1}$, the hidden state $h_t$ and the latent variable $z_t$.
2. $p_\theta(a_{t-1} \mid h_{t-1}, z_t)$ is the action decoder distribution conditioned on the the hidden states $h_{t-1}$ and the latent variable $z_t$.
3. $p_\theta(z_t \mid h_{t-1})$ is the prior over latent variable $z_t$ condition on the hidden states $h_{t-1}$

All these conditional distributions, listed above, are represented by simple distributions such as Gaussian distributions. Their means and standard variations are computed by multi-layered feed-forward networks. Although each single distribution is unimodal, the marginalization over sequence of latent variables makes $p_\theta(o_{1:T}, a_{1:T} \mid o_0)$ highly multimodal. Note that the prior distribution of the latent random variable at time step $t$ depends on all the preceding inputs via the hidden state $h_{t-1}$. This temporal structure of the prior has been shown to improve the representational power (Chung et al., 2015; Fraccaro et al., 2016; Goyal et al., 2017) of the latent variable.

## 2.2 INFERENCE MODEL

In order to overcome the intractability of posterior inference of latent variables given observation-action sequence, we make use of amortized variational inference ideas (Kingma & Welling, 2013). We consider recognition or inference network, a neural network which approximates the intractable posterior. The true posterior of a given latent variable $z_t$ is $p(z_t \mid h_{t-1}, a_{t-1:T}, o_{t:T}, z_{t+1:T})$. For the sake of an efficient posterior approximation, we make the following design choices:

1. We drop the dependence of the posterior on actions $a_{t-1:T}$ and future latent variables $z_{t+1:T}$.
2. To take into account the dependence on $h_{t-1}$, we share parameters between the generative model and the recognition model by making the approximate posterior, a function of the hidden state $h_{t-1}$ computed by the LSTM transition module $f$ of the generative model.
3. To take into account the dependence on future observations $o_{t:T}$, we use an LSTM that processes observation sequence backward as $b_t = g(o_t, b_{t+1})$, where $g$ is a deterministic transition function and $b_t$ is the LSTM backward hidden state at time $t$.

4. Finally, a feed-forward network takes as inputs $h_{t-1}$ and $b_t$ and output the mean and the standard deviation of the approximate posterior $q_\phi(z_t \mid h_{t-1}, b_t)$.

In principle, the posterior should depend on future actions. To take into account the dependence on future actions as well as future observations, we can use the LSTM that processes the observation-action sequence in backward manner. In pilot trials, we conducted experiments with and without the dependencies on actions for the backward LSTM and we did not notice a noticeable difference in terms of performance. Therefore, we chose to drop the dependence on actions in the backward LSTM to simplify the code.

Now using the approximate posterior, the Evidence Lower Bound (ELBO) is derived as follows:

$$\log p_\theta(o_{1:T}, a_{1:T} \mid o_0, h_0) \geq \mathbb{E}_{q_\phi(z_{1:T}\mid o_{0:T}, a_{0:T})}\Big[\frac{\log p_\theta(o_{1:T}, a_{1:T}, z_{1:T} \mid o_0, h_0)}{\log q_\phi(z_{1:T} \mid o_{0:T}, a_{0:T})}\Big] \tag{3}$$

$$= \mathbb{E}_{q_\phi(z_{1:T}\mid o_{0:T}, a_{0:T})}\Big[\log p_\theta(o_{1:T}, a_{1:T} \mid o_0, h_0, z_{1:T})\Big] \tag{4}$$
$$- \mathbb{KL}(q_\phi(z_{1:T} \mid o_{0:T}, a_{0:T})\|p_\theta(z_{1:T} \mid o_0, h_0))$$

Leveraging temporal structure of the generative and inference network, the ELBO breaks down as:

$$\mathcal{L}(o_{1:T}, a_{1:T}; \theta, \phi) = \sum_t \mathbb{E}_{q_\phi(z_t\mid h_{t-1}, b_t)}\Big[\log p_\theta(o_t \mid a_{t-1}, h_{t-1}, z_t) + \log p_\theta(a_{t-1} \mid h_{t-1}, z_t)\Big] \tag{5}$$

$$- \mathbb{KL}(q_\phi(z_t \mid h_{t-1}, b_t)\|p_\theta(z_t \mid h_{t-1}))$$

## 2.3 Auxiliary Cost

The main difficulty in latent variable models is how to learn a meaningful latent variables that capture high level abstractions in underlying observed data. It has been challenging to combine powerful autoregressive observation decoder with latent variables in a way to make the latter carry useful information (Chen et al., 2016; Bowman et al., 2015). Consider the task of learning to navigate a building from raw images. We try to build an internal model of the world from observation-action trajectories. This is a very high-dimensional and highly redundant observation space. Intuitively, we would like that our latent variables capture an abstract representation describing the essential aspects of the building's topology needed for navigation such as object locations and distance between rooms. The decoder will then encode high frequency source of variations such as objects' texture and other visual details. Training the model by maximum likelihood objective is not sensitive to how different level of information is encoded. This could lead to two bad scenarios: either latent variables are unused and the whole information is captured by the observation decoder, or the model learns a stationary auto-encoder with focus on compressing a single observation (Karl et al., 2016).

The shortcomings, described above, are generally due to two main reasons: the approximate posterior provides a weak signal or the model focuses on short-term reconstruction. In order to address the latter issue, we enforce our latent variables to carry useful information about the future observations in the sequence. In particular, we make use of the so-called "Z-forcing" idea (Goyal et al., 2017): we consider training a conditional generative model $p_\zeta(b \mid z)$ of backward states $b$ given the inferred latent variables $z \sim q_\theta(z \mid h, b)$. This model is trained by log-likelihood maximization:

$$\max_\zeta \mathbb{E}_{q_\theta(z\mid b, h)}[\log p_\zeta(b \mid z)] \tag{6}$$

The loss above will act as a training regularization that enforce latent variables $z_t$ to encode future information.

## 2.4 Model training

The training objective is a regularized version of the ELBO. The regularization is imposed by the auxiliary cost defined as the reconstruction term of the additional backward generative model. We bring together the ELBO in (5) and the reconstruction term in (6), multiplied by the trade-off pa-

rameter $\beta$, to define our final objective:

$$\mathcal{L}(o_{1:T}, a_{1:T}; \theta, \phi, \zeta) = \sum_t \mathbb{E}_{q_\phi(z_t|h_{t-1},b_t)}\Big[\log p_\theta(o_t \mid a_{t-1}, h_{t-1}, z_t) + \log p_\theta(a_{t-1} \mid h_{t-1}, z_t)$$

$$(7)$$

$$+ \beta \log p_\zeta(b_t \mid z_t)\Big] - \mathbb{KL}(q_\phi(z_t \mid h_{t-1}, b_t) \| p_\theta(z_t \mid h_{t-1}))$$

We use the reparameterization trick (Kingma & Welling, 2013; Rezende et al., 2014) and a single posterior sample to obtain unbiased gradient estimators of the ELBO in (7). As the approximate posterior should be agnostic to the auxiliary task assigned to the latent variable, we don't account for the gradients of the auxiliary cost with respect to backward network during optimization (7).

## 3 USING THE MODEL FOR SEQUENTIAL TASKS

Here we explain how we can use our dynamics model to help solve sequential RL tasks. We consider two settings: imitation learning, where a learner is asked to mimic an expert and reinforcement learning, where an agent aims at maximizing its long-term performance.

### 3.1 USING THE MODEL FOR IMITATION LEARNING

We consider a passive approach of imitation learning, also known as behavioral cloning (Pomerleau, 1991). We have a set of training trajectories achieved by an expert policy. Each trajectory consists of a sequence of observations $o_{1:T}$ and a sequence of actions $a_{1:T}$ executed by an expert. The goal is to train a learner to achieve – given an observation – an action as similar as possible to the expert's. This is typically accomplished via supervised learning over observation-action pairs from expert trajectories. However, this assumes that training observation-action pairs are i.i.d. This critical assumption implies that the learner's action does not influence the distribution of future observations upon which it acts. Moreover, this kind of approach does not make use of full trajectories we have at our disposals and chooses to break correlations between observation-actions pairs.

In contrast, we propose to leverage the temporal coherence present in our training data by training our dynamic model using full trajectories. The advantage of our method is that our model would capture the training distribution of sequences. Therefore, it is more robust to compounding error, a common problem in methods that fit one-step decisions.

### 3.2 USING THE MODEL FOR REINFORCEMENT LEARNING

Model-based RL approaches can be understood as consisting of two main components: (i) model learning from observations and (ii) planning (obtaining a policy from the learned model). Here, we will present how our dynamics model can be used to help solve RL problems. In particular, we explain how to perform planning under our model and how to gather data that we feed later to our model for training.

#### 3.2.1 PLANNING

Given a reward function $r$, we can evaluate each transition made by our dynamics model. A planner aims at finding the optimal action sequence that maximizes the long-term return defined as the expected cumulative reward. This can be summarized by the following optimization problem: $\max_{a_{1:T}} \mathbb{E}[\sum_{t=1}^T r_t]$ where the expectation is over trajectories sampled under the model.

If we optimize directly on actions, the planner may output a sequence of actions that induces a different observation-action distribution than seen during training and end up in regions where the model may capture poorly the environment's dynamics and make prediction errors. This training/test distribution mismatch could result in 'catastrophic failure', *e.g.* the planner may output actions that perform well under the model but poorly when executed in the real environment.

To ensure that the planner's solution is grounded in the training manifold, we propose to perform planning over latent variables instead of over actions: $\max_{z_{1:T}} \mathbb{E}[\sum_{t=1}^T r_t]$. In particular, we use model predictive control (MPC) (Mayne et al., 2000) as planner in latent space as shown in Alg. 1.

Given, an episode of length $T$, we generate a bunch of sequences starting from the initial observation, We evaluate each sequence based on their cumulative reward and we take the best sequence. Then we pick the $k$ first latent variables $z_{1:k}$ for the best sequence and we execute $k$ actions $a_{1:k}$ in the real environment conditioned on the picked latent variables. Now, we re-plan again by following the same steps described above starting at the last observation of the generated segment. Note that for an episode of length $T$, we re-plan only $T/k$ times because we generate a sequence of $k$ actions after each plan.

### 3.2.2 DATA GATHERING PROCESS

Now we turn out to our approach to collect data useful for model training. So far, we assumed that our training trajectories are given and fixed. As a consequence, the learned model capture only the training distribution and relying on this model for planning will compute poor actions. Therefore, we need to consider an exploration strategy for data generating. A naive approach would be to collect data under random policy that picks uniformly random actions. This random exploration is often inefficient in term of sample complexity. It usually wastes a lot of time in already well understood regions in the environment while other regions may be still poorly explored. A more directed exploration strategy consists in collecting trajectories that are not likely under the model distribution. For this purpose, we consider a policy $\pi_\omega$ parameterized by $\omega$ and we train it to maximize the negative regularized ELBO $\mathcal{L}$ in (7). Specifically, if $p^{\pi_\omega}(o_{1:T}, a_{1:T})$ denotes the distribution of trajectory $(o_{1:T}, a_{1:T})$ induced by $\pi_\omega$, we consider the following optimization problem:

$$\max_w \mathbb{E}_{p^{\pi_\omega}(o_{1:T}, a_{1:T})}[-\mathcal{L}(o_{1:T}, a_{1:T}; \theta, \phi, \zeta)] \qquad (8)$$

The above problem can be solved using any policy gradient method , such as proximal policy optimization PPO (Schulman et al., 2017), with negative regularized ELBO as a reward per trajectory.

The overall algorithm is described in Alg. 2. We essentially obtain a high rewarding trajectory by performs Model Predictive Control (MPC) at every $k$-steps. We then use the exploration policy $\pi_\omega$ to sample trajectories that are adjacent to the high-rewarding one obtained using MPC. The algorithm then uses the sampled trajectories for training the model.

---

**Algorithm 1** Model Predictive Control (MPC)

Given trained model M, Reward function R
**for** times $t \in \{1, ..., T/k\}$ **do**
1. Generate $m$ sequences of observation sequences of length $T_{MPC}$
2. Evaluate reward per sequence and take the best sequence.
3. Save the $k$ first latent variables $z_{1:k}$ for the best sequence (1 latent per observation)
4. Execute the actions conditioned on $z_{1:k}$ and observation $o_{1:k}$ for $k$ steps starting at the last observation of last segment.

**Algorithm 2** Overall Algorithm

Initialize replay buffer and the model with data from randomly initialized $\pi_\omega$
**for** iteration $i \in \{1, ..., N\}$ **do**
1. Execute MPC as in Algorithm 1
2. Run exploration policy starting from a random point on the trajectory visited by MPC
3. Update replay buffer with gathered data
4. Update exploration policy $\pi_\omega$ using PPO with rewards as the negative regularized ELBO
5. Train the model using a mixture of newly generated data by $\pi_\omega$ and data in the replay buffer

---

## 4 RELATED WORK

**Generative Sequence Models.** There's a rich literature of work combining recurrent neural networks with stochastic dynamics (Chung et al., 2015; Chen et al., 2016; Krishnan et al., 2015; Fraccaro et al., 2016; Gulrajani et al., 2016; Goyal et al., 2017; Guu et al., 2018). works propose a variant of RNNs with stochastic dynamics or state space models, but do not investigate their applicability to model based reinforcement learning. Previous work on learning dynamics models for Atari games have either consider learning deterministic models (Oh et al., 2015; Chiappa et al., 2017) or state space models (Buesing et al., 2018). These models are usually trained with one step ahead prediction loss or fixed k-step ahead prediction loss. Our work is related in the sense that we use stochastic RNNs where the dynamics are conditioned on latent variables, but we propose to incorporate long term future which, as we demonstrate empirically, improves over these models. In our model, the approximate posterior is conditioned on the state of the backward running RNN, which helps to escape local minima as pointed out by (Karl et al., 2016). The idea of using a bidirectional posterior

goes back to at least (Bayer & Osendorfer, 2014) and has been successfully used by (Karl et al., 2016; Goyal et al., 2017). The application to learning models for reinforcement learning is novel.

**Model based RL.** Many of these prior methods aim to learn the dynamics model of the environment which can then be used for planning, generating synthetic experience, or policy search (Atkeson & Schaal, 1997; Peters et al., 2010; Sutton, 1991). Improving representations within the context of model-based RL has been studied for value prediction (Oh et al., 2017), dimensionality reduction (Nouri & Littman, 2010), self-organizing maps (Smith, 2002), and incentivizing exploration (Stadie et al., 2015). Weber et al. (2017) introduce Imagination-Augmented Agent which uses rollouts imagined by the dynamics model as inputs to the policy function, by summarizing the outputs of the imagined rollouts with a recurrent neural network. Buesing et al. (2018) compare several methods of dynamic modeling and show that state-space models could learn good state representations that could be encoded and fed to the Imagination-Augmented Agent. Karl et al. (2017) provide a computationally efficient way to estimate a variational lower bound to *empowerement*. As their formulation assumes the availability of a differentiable model to propagate through the transitions, they train a dynamic model using Deep Variational Bayes Filter (Karl et al., 2016). (Goyal et al., 2017). (Holland et al., 2018) points out that incorporating long term future by doing Dyna style planning could be useful for model based RL. Here we are interested in learning better representations for the dynamics model using auxiliary losses by predicting the hidden state of the backward running RNN.

**Auxiliary Losses.** Several works have incorporated auxiliary loses which results in representations which can generalize. Pathak et al. (2017) considered using inverse models, and using the prediction error as a proxy for curiosity. Different works have also considered using loss as a reward which acts as a supervision for reinforcement learning problems (Shelhamer et al., 2016). Jaderberg et al. (2016) considered pseudo reward functions which helps to generalize effectively across different Atari games. In this work, we propose to use the auxillary loss for improving the dynamics model in the context of reinforcement learning.

**Incorporating the Future.** Recent works have considered incorporating the future by dynamically computing rollouts across many rollout lengths and using this for improving the policy (Buckman et al., 2018). Sutton et al. (1998) introduced TD($\lambda$), a temporal difference method in which targets from multiple time steps are merged via exponential decay. To the best of our knowledge no prior work has considered incorporating the long term future in the case of stochastic dynamics models for building better models. Many of the model based mentioned above learn global models of the system that are then used for planning, generating synthetic experience, or policy search. These methods require an reliable model and will typically suffer from modeling bias, hence these models are still limited to short horizon prediction in more complex domains (Mishra et al., 2017).

## 5 EXPERIMENTS

As discussed in Section 3, we study our proposed model under imitation learning and model-based RL. We perform experiments to answer the following questions:

1. In the imitation learning setting, how does having access to the future during training help with policy learning?
2. Does our model help to learn a better predictive model of the world?
3. Can our model help in predicting subgoals ?
4. In model-based reinforcement learning setting, how does having a better predictive model of the world help for planning and control?

### 5.1 IMITATION LEARNING

First, we consider the imitation learning setting where we have training trajectories generated by an expert at our disposal. Our model is trained as described in Section 2.4. We evaluate our model on continuous control tasks in Mujoco and CarRacing environments, as well as a partially observable 2D grid-world environments with subgoals called BabyAI (Chevalier-Boisvert & Willems, 2018).

We compare our model to two baselines for all imitation learning tasks: a *recurrent policy*, an LSTM that predicts only the action $a_t$ given an observation $o_t$, and a *recurrent decoder*, an LSTM that predicts both action and next observation given an observation. We compare to the recurrent policy to demonstrate the value of modeling future at all and we compare to the recurrent decoder to demon-

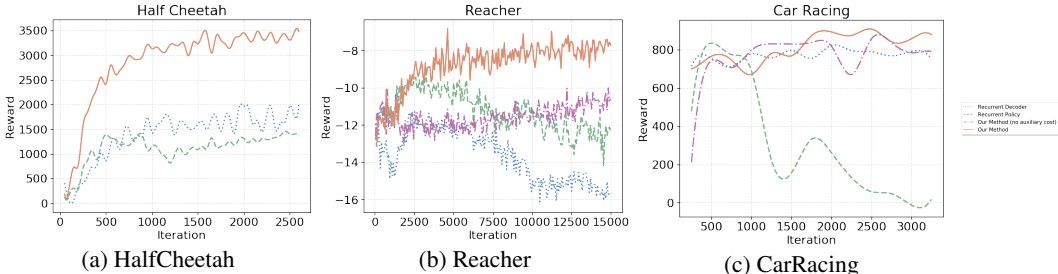

(a) HalfCheetah  (b) Reacher  (c) CarRacing

Figure 2: **Imitation Learning.** We show comparison of our method with the baseline methods for **Half-Cheetah, Reacher and Car Racing** tasks. We find that our method is able to achieve **higher reward faster** than baseline methods and is more **stable**.

strate the value of modeling long-term future trajectories (as opposite to single-step observation prediction. For all tasks, we take high-dimensional rendered image as input (compared to low-dimensional state vector). All models are trained on 10k expert trajectories and hyper parameters used are described in Section 8.1 appendix.

**Mujoco tasks.** We evaluate the models on Reacher and HalfCheetah. We take rendered images as inputs for both tasks and we compare to recurrent policy and recurrent decoder baselines. The performance in terms of test rewards are shown in Fig. 2. Our model significantly and consistently outperforms both baselines for both Half Cheetah and Reacher.

**Car Racing task.** The Car Racing task (Klimov, 2016) is a continuous control task, details for experimental setup can be found in appendix. The expert is trained using methods in Ha & Schmidhuber (2018). The model's performance compared to the baseline is shown in Fig. 2. Our model both achieves a higher reward and is more stable in terms of test performance compared to both the recurrent policy and recurrent decoder.

**BabyAI PickUnlock Task** We evaluate on the PickUnlock task on the BabyAI platform (Chevalier-Boisvert & Willems, 2018). The BabyAI platform is a partially observable (POMDP) 2D GridWorld with subgoals and language instructions for each task. We remove the language instructions since language-understanding is not the focus of this paper. The PickUnlock task consists of 2 rooms separated by a wall with a key, there is a key in the left room and a target in the right room. The agent always starts in the left room and needs to first find the key, use the key to unlock the door to go into the next room to reach to the goal. The agent receives a reward of 1 for completing the task under a fixed number of steps and gets a small punishment for taking too many steps for completing the task. Our model consistently achieves higher rewards compared to the recurrent policy baseline as shown in Fig. 3.

## 5.2 Long Horizon video prediction

One way to check if the model learns a better generative model of the world is to evaluate it on long-horizon video prediction. We evaluate the model in the CarRacing environment (Klimov, 2016). We evaluate the likelihood of these observations under the models trained in Section 5.1 on 1000 test trajectories generated by the expert trained using Ha & Schmidhuber (2018). Our method significantly outperforms the recurrent decoder by achieving a negative log-likelihood (NLL) of $-526.0$ whereas the recurrent decoder achieves an NLL of $-352.8$. We also generate images (videos) from the model by doing a 15-step rollout and the images. The video can be found at the anonymous link for our method and recurrent decoder. Note that the samples are random and not cherry-picked. Visually, our method seems to generate more coherent and complicated scenes, the entire road with some curves (not just a straight line) is generated. In comparison, the recurrent decoder turns to generated non-complete road (with parts of it missing) and the road generated is often straight with no curves or complications.

## 5.3 Subgoal detection

Intuitively, a model should become sharply better at predicting the future (corresponding to a steep reduction in prediction loss) when it observes and could easily reach a 'marker' corresponding to

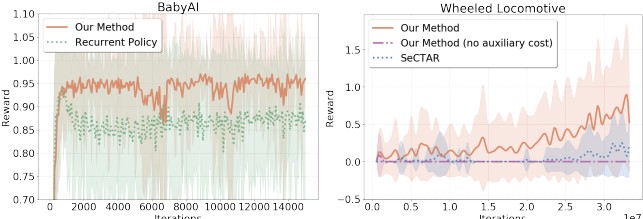

Figure 3: **Model-Based RL**. We show our comparison of our methods with baseline methods including SeCTAr for **BabyAI PickUnlock** task and **Wheeled locomotion** task with sparse rewards. We observe that our baseline achieves **higher rewards** than the corresponding baselines.

a subgoal towards the final goal. We study this for the BabyAI task that contains natural subgoals such as locating the key, getting the key, opening the door, and finding the target in the next room. Experimentally, we do indeed observe that there is sharp decrease in prediction error as the agent locates a subgoal. We also observe that there is an increase in prediction cost when it has a difficulty locating the next subgoal (no key or goal in sight). Qualitative examples of this behavior are shown in Appendix Section 8.2.

### 5.4 MODEL-BASED PLANNING

We evaluate our model on the wheeled locomotion tasks as in (Co-Reyes et al., 2018) with sparse rewards. The agent is given a reward for every third goal it reached. we compare our model to the recently proposed Sectar model (Co-Reyes et al., 2018). We outperform the Sectar model, which itself outperforms many other baselines such as Actor-Critic (A3C) (Mnih et al., 2016), TRPO (Schulman et al., 2015), Option Critic (Bacon et al., 2017), FeUdal (Vezhnevets et al., 2017), VIME (Houthooft et al., 2016) . We use the same sets of hyperparameters as in Co-Reyes et al. (2018).

## 6 CONCLUSION

In this work we considered the challenge of model learning in model-based RL. We showed how to train, from raw high-dimensional observations, a latent-variable model that is robust to compounding error. The key insight in our approach involve forcing our latent variables to account for long-term future information. We explain how we use the model for efficient planning and exploration. Through experiments in various tasks, we demonstrate the benefits of such a model to provide sensible long-term predictions and therefore outperform baseline methods.

## 7 ACKNOWLEDGEMENTS

The authors acknowledge the important role played by their colleagues at Facebook AI Research throughout the duration of this work. We are also grateful to the reviewers for their constructive feedback which helped to improve the clarity of the paper. NRK is thankful to and Nikita Kitaev and Hugo Larochelle for useful discussions. AG is thankful to Alessandro Sordoni, Sergey Levine for useful discussions. Anirudh Goyal is grateful to NSERC, CIFAR, Google, Samsung, Nuance, IBM, Canada Research Chairs, Canada Graduate Scholarship Program, Nvidia for funding, and Compute Canada for computing resources.

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

# 8 APPENDIX

## 8.1 EXPERIMENTAL SETUP

We perform the same hyper parameters search for the baseline as well as our methods. We use the Adam optimizer Kingma & Ba (2014) and tune learning rates using $[1e-3, 5e-4, 1e-4, 5e-5]$. For the hyper parameters specific for our model, we tune KL starting weight between $[0.15, 0.2, 0.25]$, the KL weight increase per iteration is fixed at $0.0005$ and the auxiliary cost for predicting the backward hidden state $b_t$ is kept at $0.0005$ for all experiments. We list the details for each experiment and task below.

**Mujoco Tasks**   We evaluate on 2 Mujoco tasks (Todorov et al., 2012), the Reacher and the Half Cheetah task(Todorov et al., 2012). The Reacher tasks is an object manipulation task consist of manipulating a 7-DoF robotic arm to reach the goal, the agent is rewarded for the number of objects it reaches within a fixed number of steps. The HalfCheetah task is continuous control task where the agent is awarded for the distance the robots moves.

For both tasks, the experts are trained using Trust Region Policy Optimization (TRPO) (Schulman et al., 2015). We generate 10k expert trajectories for training the student model, all models are trained for 50 epochs. For the HalfCheetah task, we chunk the trajectory (1000 timesteps) into 4 chunks of length 250 to save computation time.

**Car Racing task**   The Car Racing task (Klimov, 2016) is a continuous control task where each episode contains randomly generated trials. The agent (car) is rewarded for visiting as many tiles as possible in the least amount of time possible. The expert is trained using methods in (Ha & Schmidhuber, 2018). We generate 10k trajectories from the expert. For trajectories of length over 1000, we take the first 1000 steps. Similarly to Section 5.1, we chunk the 1000 steps trajectory into 4 chunks of 250 for computation purposes.

**BabyAI**   The BabyAI environment is a POMDP 2D Minigrid envorniment (Chevalier-Boisvert & Willems, 2018) with multiple tasks. For our experiments, we use the PickupUnlock task consistent of 2 rooms, a key, an object to pick up and a door in between the rooms. The agent starts off in the left room where it needs to find a key, it then needs to take the key to the door to unlock the next room, after which, the agent will move into the next room and find the object that it needs to pick up. The rooms can be of different sizes and the difficulty increases as the size of the room increases. We train all our models on room of size 15. It is not trivial to train up a reinforcement learning expert on the PickupUnlock task on room size of 15. We use curriculum learning with PPO (Schulman et al., 2017) for training our experts. We start with a room size of 6 and increase the room size by 2 at each level of curriculum learning.

We train the LSTM baseline and our model both using imitation learning. The training data are 10k trajectories generated from the expert model. We evaluate the both baseline and our model every 100 iterations on the real test environment (BabyAI environment) and we report the reward per episode. Experiments are run 5 times with different random seeds and we report the average of the 5 runs.

**Wheeled locomotion**   We use the Wheeled locomotion with sparse rewards environment from (Co-Reyes et al., 2018). The robot is presented with multiple goals and must move sequentially in order to reach each reward. The agent obtains a reward for every 3 goal it reaches and hence this is a task with sparse rewards. We follow similar setup to (Co-Reyes et al., 2018), the number of explored trajectories for MPC is 2048, MPC re-plans at every 19 steps. However, different from (Co-Reyes et al., 2018), we sample latent variables from our sequential prior which depends on the summary of the past events $h_t$. This is in comparison to (Co-Reyes et al., 2018), where the prior of the latent variables are fixed. Experiments are run 3 times and average of the 3 runs are reported.

## 8.2 CORRELATION BETWEEN SUBGOAL AND PREDICTION LOSS

Our model has an auxiliary cost associated with predicting the long term future. Intuitively, the model is better at predicting the long term future when there is more certainty about the future. Let's consider a setting where the task is in a POMDP environment that has multiple subgoals, for

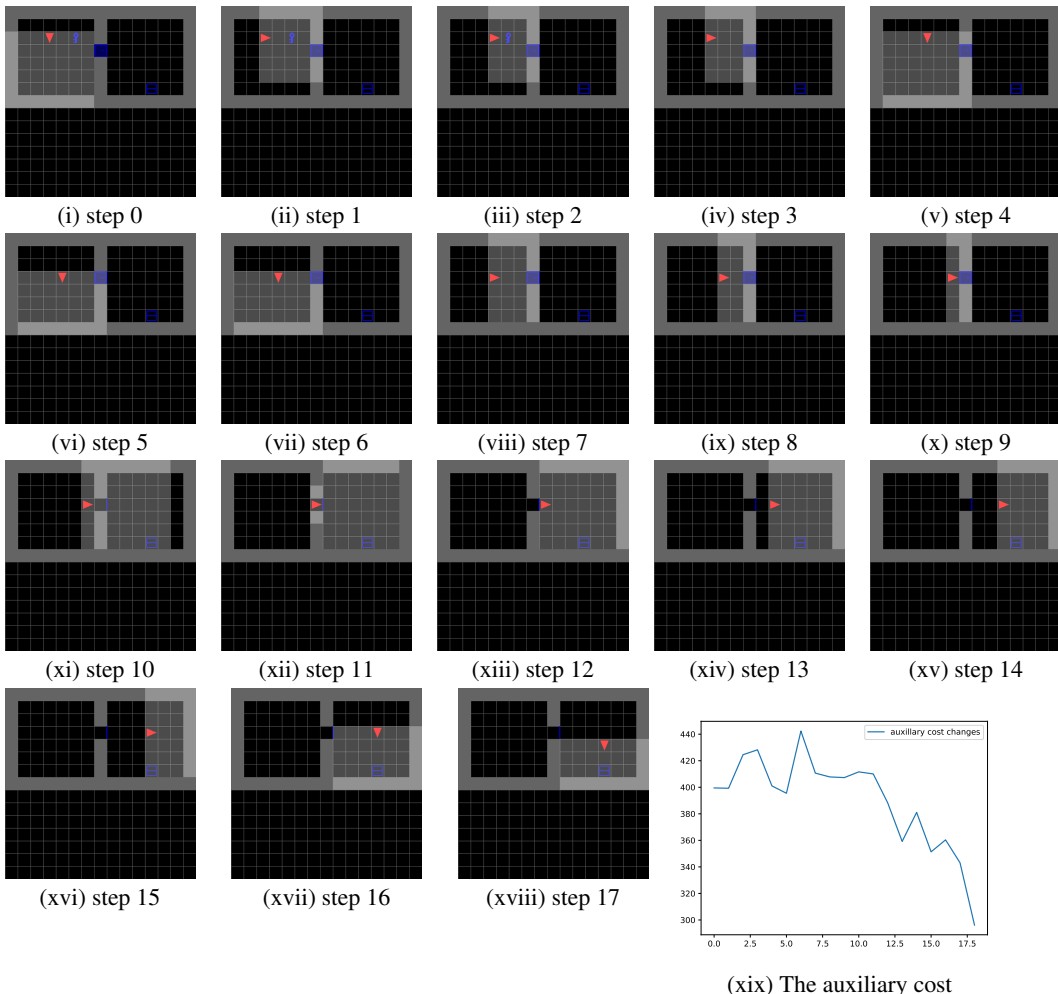

Figure 4: The first 18 plots show how the agent evolves in BabyAI environment for 18 steps. The last plot shows the the corresponding auxiliary cost in function of steps. The agent is in red. The gray regions in images are the agent's observational space. The keys are doors can be an arbitary color, in this example, both the key and the door are in blue. The auxillary cost generally descreases over time.

example the BabyAI environment (Chevalier-Boisvert & Willems, 2018) we used earlier. Intuitively, the agent or model should be more certain about the long term future when it sees a subgoal and knows how to get there and less certain if it does not have the next subgoal in sight. We test our hypothesis on tasks in the 5.1 environment.

We took our model trained using imitation learning as in section 5.1. Here, we give one example of how our model trained using imitation learning in section 5.1 behaves in real environment and how this corresponds to increase or decrease in auxiliary cost (uncertainty) described in 2.3. In figure 4, we show how our model behaves in BaybyAI environment. The last figure in 4 plots the auxiliary cost at each step. Overall, the auxiliary cost decreases as the agent moves closer to the goal and sometimes there is a sharp drop in the auxiliary cost when the agent sees the subgoal and the goal is aligned with the agent's path. An example reflecting this scenario is the sharp drop in auxiliary cost from step 6 to step 7, where the agent's path changed to be aligned with the door.

