# OpenReview forum: "Modeling the Long Term Future in Model-Based Reinforcement Learning"
_ICLR.cc/2019/Conference_

### Official Review · AnonReviewer1 · 2018-10-18
**Good idea, good paper, needs more experiment for more conclusive results**

**Rating:** 7
**Confidence:** 4

**Review:**

After the rebuttal and the authors providing newer experimental results, I've increased my score. They have addressed both the issue with the phrasing of the auxiliary loss, which I'm very happy they did as well as provided more solid experimental results, which in my opinion make the paper strong enough for publication.

#####
The paper proposes a variational framework for learning a Model of both the environment and the actor's policy in Reinforcement Learning. Specifically, the model is a deterministic RNN which at every step takes as input also a new stochastic latent variable z_t. Compared to more standard approaches, the prior over z_t is not standard normal but depends on the previously hidden state. The inference model combines information from the forward generative hidden state and a backward RNN that looks only at future observations. Finally, an auxiliary loss is added to the model that tries to predict the future states of the backward RNN using the latent variable z_t.  The idea of the paper is quite well presented and concise.

The paper tests the proposed framework on several RL benchmarks. Using it for imitation learning outperforms two baseline models: behaviour cloning and behaviour cloning trained with an auxiliary loss of predicting the next observation. Although the results are good, it would have been much better if there was also a comparison against a Generative model (identical to the one proposed) without the auxiliary loss added? The authors claim that the results of the experiment suggest that the auxiliary loss is indeed helping, where I find the evidence unconvincing given that there is no comparison against this obvious baseline. Extra comparison against the method from [1] or GAIL would make the results even stronger, but it is understandable that one can not compare against everything, hence I do not see this as a major issue.
The authors also compare on long-horizon video prediction. Although their method outperforms the method proposed in Ha & Schmidhuber, this by no means suggests that the method is really that superior. I would argue that in terms of future video prediction that [3] provides significantly better results than the World Models, nevertheless, at least one more baseline would have supported the authors claims much better.
On the Model-Based planning, the authors outperform SeCTAR model on the BabyAI tasks and the Wheeled locomotion. This result is indeed interesting and shows that the method is viable for planning. However, given that similar result has been shown in [1] regarding the planning framework it is unclear how novel the result is.

In conclusion, the paper presents a generative model for training a model-based approach with an auxiliary loss. The results look promising, however, stronger baselines and better ablation of how do different components actually contribute would make the paper significantly stronger than it is at the moment. Below are a few further comments on some specific parts of the paper.

A few comments regarding relevant literature:

Both in the introduction and during the main text the authors have not cited [1] which I think is a very closely related method. In this work similarly, a generative model of future segments is learned using a variational framework. In addition, the MPC procedure that the authors present in this paper is not novel, but has already been proposed and tried in [1] - optimizing over the latent variables rather than the actions directly, and there have been named Latent Action Priors.

The data gathering process is also not a new idea and using the error in a dynamics model for exploration is a well-known method, usually referred to as curiosity, for instance see [2] and some of the cited papers as Pathak et. al., Stadie et. al. - these all should be at least cited in section 3.2.2 as well not only in the background section regarding different topics.


On the auxiliary loss:

The authors claim that they train the auxiliary loss using Variational Inference, yet they drop the KL term, which is "kinda" an important feature of VI. Auxiliary losses are well understood that often help in RL, hence there is no need to over-conceptualize the idea of adding the extra term log p(b|z) as a VI and then doing something else. It would be much more clear and concise just to introduce it as an extra term and motivate it without referring to the VI framework, which the authors do not use for it (they still use it for the main generative model). The only way that this would have been acceptable if the experiment section contained experiments with the full VI objective as equation (6) suggest and without the sharing of the variational priors and posteriors and compared them against what they have done in the current version of the manuscript.


A minor mistake seems to be that equation (5) and (7) have double counted log p(z_t|h_t-1) since they are written as an explicit term as well as they appear in the KL(q(z_t|..)|p(z_t|h_t-1)).



[1] Prediction and Control with Temporal Segment Models [Nikhil Mishra, Pieter Abbeel, Igor Mordatch, 2017]

[2] Large-Scale Study of Curiosity-Driven Learning [Yuri Burda, Harri Edwards, Deepak Pathak, Amos Storkey, Trevor Darrell, Alexei A. Efros, 2018]

[3] Action-Conditional Video Prediction using Deep Networks in Atari Games [Junhyuk Oh, Xiaoxiao Guo, Honglak Lee, Richard Lewis, Satinder Singh, 2015]

---

> ### Author Response · Authors · 2018-11-22
> **Thanks for feedback! (1/2)**
>
> We thank the reviewer for such a detailed feedback. We have conducted additional experiments to address the concerns raised about the evaluation, and we clarify specific points below. We believe that these additions address all of your concerns about the work, though we would appreciate any additional comments or feedback that you might have. We acknowledge that the paper was certainly lacking polish and accept that this may have made the paper difficult to read in places. We have uploaded a revised version in which we have revised the problem statement and writing as per the reviewer's suggestions. We briefly summarize the key idea of the paper and then address the specific concerns.
>
> Q: “it would have been much better if there was also a comparison against a Generative model (identical to the one proposed) without the auxiliary loss added? “
>
> We thank the reviewer for pointing this out. We have updated our paper with comparisons to a generative model identical to the one we proposed but without the auxiliary cost being added. We ran comparisons against the proposed baseline (our model without auxiliary cost) both on imitation learning (Reacher, CarRacing) and RL (wheeled locomotion). Our method outperforms the baseline in all cases.
>
> Q: “However, given that similar result has been shown in [1] regarding the planning framework it is unclear how novel the result is. “
>
> The reviewer is right. We are not suggesting the use of the proposed method is novel for planning. The novelty comes from using bidirectional inference network and using the auxiliary cost for exploration. We showed that the method learns a better inference network by using the proposed method for planning, and showing that the proposed method outperforms more complicated and state of the art methods like [1], [2].
>
> [1] Sectar. https://arxiv.org/abs/1806.02813
> [2] Learning and Querying Generative Models for RL https://arxiv.org/abs/1802.03006

---

> > ### Author Response · Authors · 2018-11-22
> > **Comparison with state of the art (2/2)**
> >
> > Q: “Both in the introduction and during the main text the authors have not cited [1] which I think is a very closely related method. In this work similarly, a generative model of future segments is learned using a variational framework. In addition, the MPC procedure that the authors present in this paper is not novel, but has already been proposed and tried in [1] - optimizing over the latent variables rather than the actions directly, and there have been named Latent Action Priors. “
> >
> > We again agree with the reviewer that the paper [1] should be cited and discussed. We think, that a more related paper to our proposed method is the use of state space models where you are actually learning the dynamics model at some higher level of abstraction. We don’t claim that using the proposed method for MPC planning is novel, only the choice of bidirectional inference network and thereby leveraging variational methods and autoregressive models (RNNs) to improve training of the predictive model (at some higher level of hierarchy) in order to more accurately predict the future. Hence, use of inference network as well as using the auxiliary cost is novel (as shown by our results). We outperform both the Sectar [1] paper and learning to query paper. [2].
> >
> > [1] Sectar. https://arxiv.org/abs/1806.02813
> > [2] Learning and Querying Generative Models for RL https://arxiv.org/abs/1802.03006
> >
> >
> > Q: “The data gathering process is also not a new idea and using the error in a dynamics model for exploration is a well-known method, usually referred to as curiosity, for instance see [2] and some of the cited papers as Pathak et. al., Stadie et. al. - these all should be at least cited in section 3.2.2 as well not only in the background section regarding different topics. “
> >
> > We thank the reviewer for pointing this out. Indeed, using the error in a dynamics model for exploration is not a novel idea. The novelty of our paper comes from leveraging variational methods and autoregressive models (RNNs) to improve training of the predictive model in order to more accurately predict the future.
> >
> > Q: “The authors claim that they train the auxiliary loss using Variational Inference, yet they drop the KL term, which is "kinda" an important feature of VI. Auxiliary losses are well understood that often help in RL, hence there is no need to over-conceptualize the idea of adding the extra term log p(b|z) as a VI and then doing something else. It would be much more clear and concise just to introduce it as an extra term and motivate it without referring to the VI framework, which the authors do not use for it (they still use it for the main generative model). The only way that this would have been acceptable if the experiment section contained experiments with the full VI objective as equation (6) suggest and without the sharing of the variational priors and posteriors and compared them against what they have done in the current version of the manuscript. “
> >
> > We thank the reviewer for pointing this out. We agree with the reviewer and have updated our paper to reflect this change in Section 2.3 of the paper.
> >
> > "Comparison with state of the art state space models"
> >
> >  We first compare the proposed method to state of the art state space model  (Buesing, Lars, et al) [1].  We also note that [1] does not have an open-source implementation, and they ([1]) only evaluated on few atari games using millions of samples per game. We believe that comparing to such a strong baseline is very important and hence we compared to this on a challenging image based mujoco env, and MS_PACMAN from ALE.
> > We ask the reviewer to see the headline "Comparison with state of the art state space model - ALL REVIEWERS " for more details.
> >
> > [1] Buesing, Lars, et al. "Learning and Querying Fast Generative Models for Reinforcement Learning." *arXiv preprint arXiv:1802.03006* (2018).

---

> > > ### Author Response · Authors · 2018-11-23
> > > **Comparison with Temporal Segment Models**
> > >
> > > As the reviewer requested, we also compared the proposed method to the Temporal segment models.  We also note that this paper  does not have an open-source implementation, so we are trying to get the proposed baseline right.
> > >
> > > In order to show that the proposed model, learns a better predictive model, we first train the proposed model and the baseline using trajectories sampled from an expert policy. We then evaluate the log-likelihood on the test trajectories.
> > >
> > > Method                                                                                         Likelihood
> > > Variational  RNN					                                            1.27
> > > Prediction and Control						                    1.61
> > > Proposed model                                                                              1.84
> > >
> > > (Higher is better)
> > >
> > > This shows that the proposed method performs better as compared to the Prediction and Control baseline.
> > >
> > > We would appreciate it if the reviewer could take another look at our changes and additional results, and let us know if the reviewer would like to request additional changes that would alleviate reviewers concerns. We hope that our updates to the manuscript address the reviewer's concerns about clarity, and we hope that the discussion above addresses the reviewer's concerns about empirical significance. We once again thank the reviewer for the feedback of our work.

---

> ### Author Response · Authors · 2018-11-25
> **Request for feedback.**
>
> Thank you again for the thoughtful review. We would like to know if our rebuttal  adequately addressed your concerns. We would also appreciate any additional feedback on the revised paper.  (We have compared to the Learning to query paper and Prediction and Control paper which the reviewer asked, and also rewritten the motivation behind the KL term).
> Are there any other aspects of the paper that you think could be improved?

---

> > ### Author Response · Authors · 2018-11-29
> > **Feedback very useful!**
> >
> > Dear Reviewer,
> >
> > Your feedback has already been very helpful in improving the paper. We would like to know if our response adequately addressed your concerns.
> >
> > As  the discussion period is coming to an end. If you have any questions or would like to provide more specific context behind your scores, we would be happy to provide more feedback. Are there any other aspects of the paper that you think could be improved?

---

> > > ### Comment · AnonReviewer1 · 2018-12-08
> > > **Feedback**
> > >
> > > I want to thank the authors for the thorough engagement with the reviewers and the additional effort for improving the original submission.

---

> > ### Comment · AnonReviewer1 · 2018-11-30
> > **Feedback**
> >
> > I think the authors addressed really well the reviewes and engaged a lot on this paper with providing additional results and experiments which were requested by reviewers. I think the rebuttal was very adequare and definately would be useful for both reviewers and any outside readers. I will revise all of the provided extra information and the new draft of the paper and make a revision based on those.
> > Thank you for the great communication.

---

> > > ### Author Response · Authors · 2018-11-30
> > > **Feedback**
> > >
> > > Dear Reviewer,
> > >
> > > Thanks for encouraging words. We note that except adding comparison to the Learning to query paper, other results are already added to the paper (i.e the intuition behind the auxiliary cost as the reviewer suggested, related work section as other reviewer suggested, comparison to the baseline without auxiliary loss). Since we cant update the paper now, if the paper gets accepted , we will update it then!
> > >
> > > Thanks for your time! :)

---

### Official Review · AnonReviewer2 · 2018-11-01
**Interesting approach; not sure if really scales to long horizon problems**

**Rating:** 6
**Confidence:** 4

**Review:**

The paper introduces an interesting approach to model learning for imitation and RL. Given the problem of maintaining multi-step predictions in the context of sequential decision making process, and deficiencies faced during planning with one-step models [1][2], it’s imperative to explore approaches that do multi-step predictions. This paper combines ideas from learning sequential latent models with making multi-step future predictions as an auxiliary loss to improve imitation learning performance, efficiency of planning and finding sub-goals in a partially observed domain.

From what I understand there are quite a few components in the architecture. The generative part uses the latent variables z_t and LSTM hidden state h_t to find the factored autoregressive distribution p_\theta. It’s slightly unclear how their parameters are structured and what parameters are shared (if any). I understand these are hard to describe in text, so hopefully the source code for the experiments will be made available.

On the inference side, the paper makes a few choices to make the posterior approximation. It would be useful to describe the intuitions behind the choices especially the dependence of the posterior on actions a_{t-1}:T because it seems like the actions _should_ be fairly important for modeling the dynamics in a stochastic system.

In the auxiliary cost, it’s unclear what q(z|h) you are referring to in the primary model. It’s only when I carefully read Eq 7, that I realized that it’s p_\theta(z|h) from the generator.

Slightly unsure about the details of the imitation and RL  (MPC + PPO + Model learning) experiments. How large is the replay buffer? What’s the value of k? It would be interesting how the value of k affects learning performance. It’s unclear how many seeds experiments were repeated with.

Overall it’s an interesting paper. Not sure if the ideas really do scale to “long-horizon” problems. The MuJoCo tasks don’t need good long horizon models and the BabyAI problem seems fairly small.

- Minor points

Sec 2.3: not sensitive *to* how different
Algorithm 2: *replay* buffer

[1]: https://arxiv.org/abs/1612.06018
[2]: https://arxiv.org/abs/1806.01825

---

> ### Author Response · Authors · 2018-11-22
> **Thanks for feedback! (1/2)**
>
> We thank the reviewer for such a detailed feedback. We have conducted additional experiments to address the concerns raised about the evaluation, and we clarify specific points below. We believe that these additions address all of your concerns about the work, though we would appreciate any additional comments or feedback that you might have. We acknowledge that the paper was certainly lacking polish and accept that this may have made the paper difficult to read in places. We have uploaded a revised version in which we have added the extra references as per the reviewer's suggestions.
>
> We have conducted additional experiments to compare the proposed model to the state of the art state space model. We ask the reviewer to refer to the heading "Comparison with state of the art state space model - ALL REVIEWERS"
>
> Q: “From what I understand there are quite a few components in the architecture. The generative part uses the latent variables z_t and LSTM hidden state h_t to find the factored autoregressive distribution p_\theta. It’s slightly unclear how their parameters are structured and what parameters are shared (if any). I understand these are hard to describe in text, so hopefully the source code for the experiments will be made available.”
>
> We thank the reviewer for pointing this out. We will publish our source code. Our model consists of a forward and backward RNN, with the forward and backward RNNs do not share parameters.
>
> Q: “On the inference side, the paper makes a few choices to make the posterior approximation. It would be useful to describe the intuitions behind the choices especially the dependence of the posterior on actions a_{t-1}:T because it seems like the actions _should_ be fairly important for modeling the dynamics in a stochastic system.‘
>
> We thank the reviewer for pointing this out. In principle, the posterior should depend on future actions. To take into account the dependence on future actions as well as future observations, we can use the LSTM that processes the observation-action sequence backwards. In pilot trials, we conducted experiments with and without the dependencies on actions for the backward LSTM and we didn’t notice a noticeable difference in terms of performance. We hence chose to drop the dependencies on actions in the backward LSTM to simplify the code. We have updated the paper (appendix) to clarify this difference.

---

> > ### Author Response · Authors · 2018-11-22
> > **Comparison to the state of the art models (2/2)**
> >
> > Q: “Slightly unsure about the details of the imitation and RL  (MPC + PPO + Model learning) experiments. How large is the replay buffer? What’s the value of k? It would be interesting how the value of k affects learning performance. It’s unclear how many seeds experiments were repeated with.”
> >
> > We agree with the reviewer. We have added the details about each experiment’s setup in the appendix. For example, we use k=19 for wheeled locomotion tasks. We followed the exact same setup (hyperparameters) as SeCTAr[1] and so we did not try any other values of k to ensure fairness.  All imitation learning experiments and RL experiments are repeated 5 times with different random seeds.
> >
> > [1] Sectar. https://arxiv.org/abs/1806.02813
> >
> > Q: “Not sure if the ideas really do scale to “long-horizon” problems. The MuJoCo tasks don’t need good long horizon models and the BabyAI problem seems fairly small.”
> >
> > We thank the reviewer for pointing this out, we agree this is an important point. The wheeled locomotion tasks we used in our experiments is a challenging task with long-horizon planning. The agent is presented with multiple goals and sparse rewards. The agent needs to plan to reach each goal sequentially, only after reaching the 3rd goal, the agent obtains a reward of 1.  We currently outperform the Sectar paper which we believe is a very strong baseline. We ran other experiments to compare our proposed method with the state of the art state space models[1]. We ask the reviewer to refer to the "ALL REVIEWERS" headline.
> >
> > [1] Learning and Querying Generative Models for RL https://arxiv.org/abs/1802.03006
> >
> > We would appreciate it if the reviewer could take another look at our changes and additional results, and let us know if the reviewer would like to request additional changes that would alleviate reviewers concerns. We hope that our updates to the manuscript address the reviewer's concerns about clarity, and we hope that the discussion above addresses the reviewer's concerns about empirical significance. We once again thank the reviewer for the feedback of our work.

---

> ### Author Response · Authors · 2018-11-25
> **Request for feedback.**
>
> Thank you again for the thoughtful review. We would like to know if our rebuttal  adequately addressed your concerns. We would also appreciate any additional feedback on the revised paper.  (We have compared to the Learning to query paper and Prediction and Control paper which the reviewer asked, and added the missing baselines). Are there any other aspects of the paper that you think could be improved?

---

> > ### Author Response · Authors · 2018-11-29
> > **Feedback Useful! thanks :)**
> >
> > Dear Reviewer,
> >
> > Your feedback has already been very helpful in improving the paper. We would like to know if our response adequately addressed your concerns.
> >
> > As  the discussion period is coming to an end. If you have any questions or would like to provide more specific context behind your scores, we would be happy to provide feedback. Are there any other aspects of the paper that you think could be improved?
> >
> > thanks for your time! :)

---

### Official Review · AnonReviewer3 · 2018-11-05
**Review of "Modeling the Long Term Future in Model-Based Reinforcement Learning**

**Rating:** 6
**Confidence:** 4

**Review:**

The authors claim that long-term prediction as a key issue in model-based reinforcement learning. Based on that, they propose a fairly specific model to which is then improved with Z-forcing to achieve better performance.

## Major

The main issue with the paper is that the premise is not convincing to me. It is based on four works which (to me) appear to focus on auto-regressive models. In this submission, latent variable models are considered. The basis for sequential LVMs suffering from these problems is therefore not given by the literature.

That alone would not be much of an issue, since the problem could also be shown to exist in this context in the paper. But the way I understand the experimental section, the approach without the auxiliary cost is not even evaluated. Therefore, we cannot assess if it is that alone which improves the method. The central hypothesis of the paper is not properly tested.

Apart from that, the paper appears to have been written in haste. There are numerous typos in text and in equations (e.g. $dz$ missing from integrals).

To reconsider my assessment, I think it should be shown that the problem of long-term future prediction exists in the context of sequential LVMs. Maybe this is obvious for ppl more knowledgeable in the field, but this paper fails to make that point by either pointing out relevant references or containing the necessary experiments. Especially since other works have made model-based control work in challenging environments:

- Buesing, Lars, et al. "Learning and Querying Fast Generative Models for Reinforcement Learning." *arXiv preprint arXiv:1802.03006* (2018).
- Karl, M., Soelch, M., Becker-Ehmck, P., Benbouzid, D., van der Smagt,
  P., & Bayer, J. (2017). Unsupervised Real-Time Control through
  Variational Empowerment. *arXiv preprint arXiv:1710.05101*.

## Minor

- The authors chose to use the latent states for planning. This turns the optimisation into a POMDP problem. How is the latent state inferred at run time? How do we assure that the policy is still optimal?
- Application of learning models to RL is not novel, see references above. But maybe this is a misunderstanding on my side, as the Buesing paper is cited in the related work.

---

> ### Author Response · Authors · 2018-11-22
> **Thanks for review! Comparison against state space models. (1/3)**
>
> We thank the reviewer for such a detailed feedback. We have conducted additional experiments to address the concerns raised about the evaluation, and we clarify specific points below. We believe that these additions address all of your concerns about the work, though we would appreciate any additional comments or feedback that you might have. We acknowledge that the paper was certainly lacking polish and accept that this may have made the paper difficult to read in places. We have uploaded a revised version in which we have added the extra references as per the reviewer's suggestions.
>
> Q: “The main issue with the paper is that the premise is not convincing to me…., latent variable models are considered. The basis for sequential LVMs suffering from these problems is therefore not given by the literature.”
>
> We thank the reviewer for pointing this out. There exists a rich literature which advocates learning sequential LVMs, and mentions that its difficult to learn sequential LVMs. In particular, [1], [2], [3], [4], [5], [6], [7], [8]  observes that it is difficult extracting meaningful representations for latent variables when a powerful autoregressive model is used. We will update the paper to reflect these.
>
> [1] A recurrent latent variable model for sequential data, https://arxiv.org/abs/1506.02216
> [2] Variational Lossy Autoencoder, https://arxiv.org/abs/1611.02731
> [3] Generating sentences from a continuous space, https://arxiv.org/abs/1511.06349
> [4] Sequential neural models with stochastic layers https://arxiv.org/abs/1605.07571
> [5] A hierarchical latent variable encoder-decoder model for generating dialogues.  https://arxiv.org/abs/1605.06069
> [6] Gulrajani, I., Kumar, K., Ahmed, F., Taiga, A. A., Visin, F., Vazquez, D., and Courville, A. (2016). Pixelvae: A latent variable model for natural images. arXiv preprint arXiv:1611.05013.
> [7] Variational Autoencoders for semi supervised text classification https://arxiv.org/abs/1603.02514
> [8]. GOYAL, Anirudh Goyal ALIAS PARTH, et al. "Z-Forcing: Training stochastic recurrent networks." Advances in Neural Information Processing Systems. 2017.
>
> This was also observed in our paper, as our method without auxiliary cost noticeably under-performs compared to our method with auxiliary cost which suggests that auxiliary cost acts as an important regularizer.
>
> Q: “That alone would not be much of an issue, since the problem could also be shown to exist in this context in the paper. But the way I understand the experimental section, the approach without the auxiliary cost is not even evaluated.”
>
> We thank the reviewer for pointing this out and we agree that this is an important baseline. We ran comparisons against the proposed baseline (our model without auxiliary cost) both on imitation learning (Reacher) and RL (wheeled locomotion). Our method outperforms the baseline in all cases.
> We also ran other experiments to compare the proposed method with state of the art methods[1] like the one mentioned by the reviewer.
>
> [1] Buesing, Lars, et al. "Learning and Querying Fast Generative Models for Reinforcement Learning." *arXiv preprint arXiv:1802.03006* (2018).
>
> Q: “There are numerous typos in text and in equations (e.g. $dz$ missing from integrals).”
>
> We thank the reviewer for pointing these out. We have corrected these typos in the updated paper.

---

> > ### Author Response · Authors · 2018-11-22
> > **Long term prediction sequential LVM's (2/3)**
> >
> > " Especially since other works have made model-based control work in challenging environments:/ Application of learning models to RL is not novel, see references above. But maybe this is a misunderstanding on my side, as the Buesing paper is cited in the related work.”"
> >
> > We thank the reviewer for pointing out the references for other model-based RL works. We have updated these references in the “related works” section.  We build our work on many works which explore models ranging from deterministic recurrent neural networks (RNNs) to fully stochastic models, to be more precise on the comparisons between our work compared with other works in this area.
> >
> > [1], [2], [5] train stochastic stochastic RNN with latent variables, but not in the context of model based reinforcement learning (i.e not for building model of the environment, but for supervised or unsupervised learning learning tasks such as language modeling and speech modeling). [6], [7] train an action-conditioned video prediction network by first learning the latent representation, and then using that latent representation for learning the model. Similar to [9], we present stochastic sequence models that work on high-dimensional data.
> >
> >
> > [1] Black box variational inference for state space models. https://arxiv.org/abs/1511.07367
> > [2] Recurrent Latent Variable for sequential data https://arxiv.org/abs/1506.02216
> > [3] Sequential neural models with stochastic layers https://arxiv.org/abs/1605.07571
> > [4] Deep kalman filters. https://arxiv.org/abs/1511.05121
> > [5] Z-Forcing https://arxiv.org/abs/1711.05411
> > [6] Embed to control: A locally linear latent dynamics model for control from raw images.
> > [7] From pixels to torques: Policy learning with deep dynamical models.
> > [8] Value prediction network. https://arxiv.org/abs/1707.03497
> > [9] Learning and Querying Generative Models for RL https://arxiv.org/abs/1802.03006
> >
> >
> > Q:” The authors chose to use the latent states for planning. This turns the optimisation into a POMDP problem. How is the latent state inferred at run time? How do we assure that the policy is still optimal?
> >
> > At the run time, latent variables are inferred by executing a planning algorithm. In the case of MPC, which we use in our RL experiments, we generate multiple samples of latents variables from the prior p(z_t| h_{t-1}), we evaluate the corresponding generated trajectories and then we pick the latent variable sequence that gives the most rewarding trajectory. By planning over latent variables and not over actions directly, we assure that the actions generated by the optimal latent variables are also approximately optimal with respect to state-action distribution captured by the model. If we assume that we explore enough and that our model is accurate enough, the obtained policy is ensured to be optimized towards maximizing expected rewards.

---

> > > ### Author Response · Authors · 2018-11-22
> > > **Learning better dynamics model using auxiliary cost and bidirectional inference (3/3)**
> > >
> > > Q:” Application of learning models to RL is not novel, see references above. But maybe this is a misunderstanding on my side, as the Buesing paper is cited in the related work.”
> > >
> > > We agree that the application of learning models to RL is not novel, the novelty of our method comes from building a more accurate predictive model of the environment. Our methods differs from previous works in many aspects. Mainly, we differ on our model architecture, which dictates how to train the model and how to use it for control or sequential task in general. For instance, Buesing et al. use a pretrained state-space model to generate trajectories of latent states. These are encoded by an LSTM and the obtained embedding is fed to a policy network along with the real observations. This policy network is then trained using a model-free method. Therefore, model accuracy is not really critical in their setting as they don’t execute an explicit planning with the learned model. In our work, we focus on training a sequential LVM model that learns a better model of the longer term future by forcing latent variables to carry information about future observations. The accuracy of our model is critical in our setting in order to provide sensible explicit planning. In our experiments, we also compared the proposed method to the work of Buesing et. al, and in our preliminary experiments, we outperform as compared to their work.
> > >
> > > We would appreciate it if the reviewer could take another look at our changes and additional results, and let us know if the reviewer would like to request additional changes that would alleviate reviewers concerns. We hope that our updates to the manuscript address the reviewer's concerns about clarity, and we hope that the discussion above addresses the reviewer's concerns about empirical significance. We once again thank the reviewer for the feedback of our work.

---

> ### Author Response · Authors · 2018-11-25
> **Request for Feedback.**
>
> Thank you again for the thoughtful review. We would like to know if our rebuttal  adequately addressed your concerns. We would also appreciate any additional feedback on the revised paper. Are there any other aspects of the paper that you think could be improved?

---

> > ### Author Response · Authors · 2018-11-29
> > **Thanks for increasing score!**
> >
> > Dear Reviewer,
> >
> > We thank the reviewer for taking time in reading our feedback, and increasing their score.
> > Your feedback has already been very helpful in improving the paper.
> >
> > Thanks!

---

### Author Response · Authors · 2018-11-22
**Comparison with state of the art state space model**

We thank the reviewers for such a detailed feedback. We have conducted additional experiments to address the concerns raised about the evaluation, and we clarify specific points below. We believe that these additions address the shared concerns of the reviewers. We will address individual reviewer's concerns in their respective threads and we still welcome any additional comments or feedback that you might have.

We first compare the proposed method to state of the art state space model  (Buesing, Lars, et al) [1].  We also note that [1] does not have an open-source implementation, and they ([1]) only evaluated on few atari games using millions of samples per game. We believe that comparing to such a strong baseline is very important and hence we compared to this on a challenging image based mujoco env, and MS_PACMAN from ALE.  For our evaluations on image based mujoco domain, we use image-based continuous control tasks (half-cheetah). This environments provide qualitatively different challenges, as its nearly impossible to infer the velocity of the half-cheetah just from images, and hence using only the images makes the task partially observable (and challenging).  We compare the proposed method with the state-of-the-art Learning to Query model (Buesing, Lars, et al) [1]. In order to show that the proposed model, learns a better predictive model, we first train the proposed model and the baselines using trajectories sampled from an expert policy. We evaluate both the proposed model, and the baseline by predicting the future for longer timesteps (100 timesteps) than it was train for (50 time steps). We demonstrate that the proposed model helps to learn a better model with improved long term dependencies by making the latent variable z conditioned on the future, and using the latent variable for predicting the future.

Method                                                                                         Likelihood
Variational  RNN					                                            1.2
Learning to query						                            1.62
Proposed model without the auxiliary cost                                1.59
Proposed model                                                                              1.79


Here, we compute the likelihood of the predicted trajectories.  This result shows that the trajectories generated by the proposed model are more likely as compared to the baseline methods. We compare the proposed method to stochastic RNN, learning to query paper, proposed model (without auxiliary cost) and proposed model with auxiliary cost.

=================================

We also compare the proposed model on the ms_pacman env from the atari. This env was chosen to cover a broad range of env. dynamics. The data was collected by running a pretrained policy, and collecting sequence of observations, actions and rewards for 10 time steps. Results are computed on held out test set. Since, each of these models takes considerable amount of time to compute, we did not do any hyperparameter search neither for the baseline, nor for our proposed method. We report likelihood improvements over a baseline model.  We consider 3 baselines [1] Variational RNN [2] learning to query paper [3] proposed method without the auxiliary cost [4] Proposed method with auxiliary cost.

We report Improvement of test likelihoods of environment models over a baseline model

Method                                                                                         Likelihood (in units of 10^-3.nats/pixel)
Variational RNN								                     1.4
Learning to query						                             1.77
Proposed model                                                                               1.85
Proposed model without the auxiliary cost                                 1.69

For this env also, the proposed model outperforms both the learning to query model, as well as the baseline without the auxiliary cost.

---

> ### Comment · AnonReviewer1 · 2018-11-26
> **Clarification of the Likelihood values**
>
> Thanks again for incorporating this feedback and for the effort in improving the submition.
>
> I have a question regarding the first table in the above comment specifically:
> "Here, we compute the likelihood of the predicted trajectories.  This result shows that the trajectories generated by the proposed model are more likely as compared to the baseline methods. We compare the proposed method to stochastic RNN, learning to query paper, proposed model (without auxiliary cost) and proposed model with auxiliary cost."
>
> I'm a bit confused what exactly do you mean that you evaluate. Do you mean that you calculate:
> E[log p_model(trajectory|past)]_{tranjectory sampled according to p_model}
> or do you mean
> E[log p_model(trajectory|past)]_{tranjectory sampled from true environment}
>
> The language used (predicted trajectories) indicates the former. However, I'm not sure this indicates nessacarily that one model is better than another as it more-likely tells us what is the entropy of the distribution, but not if that distribution is good in any way.
>
> Also thanks for the MS Pacman experiments as well.

---

> > ### Author Response · Authors · 2018-11-26
> > **More clarification**
> >
> > We apologize for the confusion. We evaluate the likelihood of heldout trajectory (i.e "test set" trajectories).
> >
> >
> > We meant this, i.e we sample the trajectory from the "true" env, and then evaluate likelihood under the proposed model, and the rest of the baselines.
> > E[log p_model(trajectory|past)]_{trajectory sampled from true environment}
> >
> > We would appreciate it if the reviewer could take another look at our changes and additional results, and let us know if the reviewer would like to request additional changes that would alleviate reviewers concerns.
> >
> > Once the reviewer says yes, we would update the first reply so that other readers dont get confused. Thanks again for taking time in reading our reply.

---

> > > ### Comment · AnonReviewer1 · 2018-11-26
> > > **Clarification understood**
> > >
> > > Thanks, that makes sense. Feel free to update your original comment.

---

> > > > ### Author Response · Authors · 2018-11-27
> > > > **Feedback by reviewer**
> > > >
> > > > We would appreciate it if the reviewer could take another look at our changes and additional results, and let us know if the reviewer would like to request additional changes that would alleviate reviewers concerns, and let us know if you would like to either revise your rating of the paper.
> > > >
> > > >  We once again thank the reviewer for the feedback of our work.
> > > >
> > > > Thanks for your time! :)

---

### Author Response · Authors · 2018-11-27
**Final Rebuttal**

We would like to thank all the reviewers for taking time and giving detailed feedback on our paper.  Feedback by the reviewers have already been very helpful in improving the paper. We are also glad that the reviewers found our paper to be "quite well presented and concise." (Reviewer 1) and "an interesting paper" (Reviewer 2).

We would open source the code for the proposed method.

We conducted additional experiments, and rewrote certain sections of the paper to make it more concise.

- We conducted additional experiments and compared our paper with the state of the art state space model. (Buesing, Lars, et al. [1]). We found that in our preliminary experiments our model performs better as compared to [1]. (Reviewer 1 and Reviewer 3). We also note that the source code for this paper is not available (Buesing, Lars, et al. [1]), and the authors only evaluated their method on Atari taking millions of samples.

- We also conducted additional experiments comparing the proposed approach as to when no auxiliary cost was included (all reviewers). We find that the proposed approach performs better as compared to the case when auxiliary cost is not included (and hence showing that our model is learning a better predictive model). (ALL REVIEWERS)

- We added more references showing that the problem of long-term future prediction exists in the context of sequential LVMs. (Reviewer 3).

- We updated the part of the paper where we describe the KL cost. (Reviewer1).

- We also ran additional experiments comparing our work with the prediction and control paper as pointed by Reviewer 1. Here, also we outperform the proposed baseline. Again, we note that this paper does not have open source code base.


[1] Buesing, Lars, et al. "Learning and Querying Fast Generative Models for Reinforcement Learning." *arXiv preprint arXiv:1802.03006* (2018).

We feel that conducting additional experiments has improved the quality of the paper and we also think that we have appropriately addressed the comments by the reviewers.

We again thank all the reviewers, area chair for  their time.

Thank you! :-)

---

### Meta-Review · Area_Chair1 · 2018-12-14

**Confidence:** 4
**Recommendation:** Accept (Poster)

**Metareview:**

This paper explores the use of multi-step latent variable models of the dynamics in imitation learning, planning, and finding sub-goals. The reviewers found the approach to be interesting. The initial experiments were a main weakpoint in the initial submission. However, the authors updated the experimental results to address these concerns to a significant degree. The reviewers all agree that the paper is above the bar for acceptance. I recommend accept.